

# Micro business participation in collective flood adaptation. Lessons from scenario-based analysis in Ho Chi Minh City, Vietnam

Prof. Dr. Javier Revilla Diez, Institute of Geography, University of Cologne, Germany
j.revilladiez@uni-koeln.de, ORCID https://orcid.org/0000-0003-2065-1380

Dr. Roxana Leitold, Institute of Geography, University of Cologne, Cologne, Germany,
r.leitold@uni-koeln.de, ORCID https://orcid.org/0000-0002-2946-6774

Dr.Van Tran, University of Economics and Law, Vietnam National University, Ho Chi Minh City, Vietnam,
vantq@uel.edu.vn, ORCID https://orcid.org/0000-0002-5407-5038

Prof. Dr. Matthias Garschagen, Department of Geography, Ludwig-Maximilians-University Munich, Germany
M.Garschagen@geographie.uni-muenchen.de, https://orcid.org/0000-0001-9492-4463

**Abstract:**

Although research on the impacts of climate change on small and medium-sized enterprises (SMEs) and their adaptation to climate change risks has recently received more attention, the focus on micro and household businesses is still very limited. Micro and household businesses are adversely affected by compound flooding events - a situation that will become more acute in the future - but there is little attention in the scientific literature on their adaptation options and actual implementation. Against this background, the paper analyzes the following research questions How are micro-businesses already responding to flooding? Are micro-businesses willing to collectively invest in future proactive adaptation efforts in their neighborhood? What are the key drivers and barriers to adaptation? Based on scenario-based field experiments in Ho-Chi-Minh City, our results show that micro-businesses could play a much larger role in collective adaptation. Often overlooked in adaptation research, their willingness to engage in collective action under severe constraints is surprising. The conceptual framework presented in this paper helps us to understand the key drivers and barriers of micro-businesses' willingness to participate in collective adaptation activities. The most important key barriers for micro-businesses are limited financial capacity and lack of support from local authorities. However, micro-businesses are willing to contribute depending on the concrete adaptation measure and financing options. If no financial contribution is expected, almost 70% are willing to participate in awareness raising campaigns. And although their financial capacity is very limited, 39% of micro-businesses would contribute financially if the costs were shared with other businesses in their neighborhood and with local authorities. In this context, micro-businesses should be much more involved in adaptation plans and measures. Through their local embeddedness, they can be important multipliers in strengthening adaptive capacity at the local level.

Author contribution: Javier Revilla Diez and Matthias Garschagen develop the conceptual framework, Roxana Leitold designed the experiments and carried them out with Van Tran. Javier Revilla Diez prepared the manuscript with contributions from all co-authors.

Competing interests: The authors declare that they have no conflict of interest.




## 1 Introduction

In many countries of the Global South micro businesses together with small and medium sized businesses build up the "economic and social fabric" (Chaudhury 2018). In an urban context they include individuals or households who are shopkeepers, run cafes, restaurants, or repair shops, offer transport and warehouse or construction and maintenance services, often located in the middle of residential neighborhoods. According to the UN (2015) these businesses are a key engine of job creation and responsible for more than 50% of total employment. However, these micro, small and medium sized businesses are facing tremendous challenges in respect to climate change. A very illustrative example is Ho-Chi-Minh City (HCMC). HCMC is already facing manifold challenges due to regular flooding, which are projected to be aggravated by future climate change (Downes et al., 2016; Downes and Storch, 2014; Duy et al., 2018; Nicholls et al., 2007).

Although research on the impacts of climate change on small- and medium-sized firms (SMEs) and their adaptive behavior against climate change risks recently have received more attention (e.g., (Halkos and Skouloudis, 2019; Howe, 2011; Marks and Thomalla, 2017; Neise et al., 2019; Neise et al., 2018; Neise and Revilla Diez, 2019; Pathak and Ahmad, 2018; Pathak and Ahmad, 2016; Pauw and Chan, 2018), the focus on micro and household businesses is still very limited.

Micro businesses typically have limited financial resources to invest in both short- and long-term adaptation measures (Leitold et al., 2021; Ngin et al., 2020) and underdeveloped capabilities in business planning (Gherhes et al., 2016). However, because they bear the brunt of climate-related impacts, generate high shares of employment, and are thus closely linked to peoples' livelihoods, the discussion of the significance and prospects of micro businesses in responding to climate impacts has received attention in adaptation research (Chaudhury, 2018; Schaer et al., 2019). Crick et al. (2018) and Pulver and Benney (2013) exemplify that not all businesses have the same adaptive capacity, respond in the same way, and consider climate change as part of their business operations. What Daddi et al. (2018) and Linnenluecke et al. (2013) already illustrated for SMEs is especially true for micro businesses: Their decision-making for or against adaptation action is still underexplored and remains a black-box (Crick et al., 2018; Pauw and Chan, 2018). Recently, multi-stakeholder initiatives involving small- and medium-sized businesses have been discussed as door-openers for private sector engagement in adaptation efforts (Challies et al., 2016; Chen et al., 2013; Leitold et al., 2020; Neise et al., 2019). But, how successful can these initiatives be without exactly knowing how micro businesses are impacted and reacting to climate change, which adaptive capacities they possess and whether their adaptation behavior would change if for example incentives like financial support is provided?

Against this backdrop, this paper explores the potential role of micro-businesses in collaborative adaptation initiatives. We will focus on the following research questions: How do micro firms already respond to flooding? And more future oriented, under which conditions are micro firms willing to invest jointly into proactive adaptation efforts in their neighborhood?

Our methodological approach is twofold. First, by using scenario-based field experiments we examine the willingness of micro businesses to invest in collective adaptation options depending on different financing options. We analyze how the distribution of costs among other micro businesses and the neigbourhood, or financial incentives provided by local authorities, or pure political pressure impacts the willingness of micro businesses to contribute financially to different adaptation scenarios like the implementation of a dike system, a drainage system or awareness programs. Second, we applied a two-level binary-logistic regression that allows us to consider the differences and interdependencies between adaptation scenario and micro business characteristics in order to the detect the key drivers and barriers for adaptation. The necessary data was generated during a household and micro business survey as part of a collaborative research project "DECIDER" (Decisions for the Design of Adaptation



Pathways and the Integrative Development, Evaluation, and Governance of Flood Risk Reduction
Strategies in Changing Urban-Rural Systems). A total of 252 micro businesses were surveyed in HCMC
between September and November 2020. In addition, we were able to conduct the scenario-based
experiments with 62 out of the 252 micro businesses. As each participant responded to 20 scenarios
1,240 observations were generated for data processing.
This article is organized as follows. Section 2 develops a conceptual framework on potential drivers
and barriers of micro business adaptation. Section 3 introduces the study area and methodology.
Section 4 presents the descriptive and analytical results of the study, while Section 5 discusses the
implications of the results for addressing micro business perspective in collective flood adaptation.
Section 6 provides a conclusion.

## 104 2    Conceptual considerations
### 105 2.1 What do we know from adaptation literature?

Businesses play important roles in economic and social development worldwide by providing
employment, goods, value added, services, and taxes (Halkos et al., 2018; Leitold et al., 2020; Lo et al.,
2019). However, the fifth IPCC Assessment Report (2014) revealed a striking gap in existing scientific
literature on private sector adaptation to floods (e.g., Berkhout et al., 2006; Linnenluecke et al., 2013;
Linnenluecke et al., 2011). Since then, a body of literature has emerged that focuses on large and
multinational enterprises, that are understood to be important entities for financing adaptation
projects, developing technologies, and innovative adaptation solutions (Averchenkova et al., 2016;
Haraguchi and Lall, 2015; Neise et al., 2018). However, this focus on large, international enterprises
provides only limited knowledge on adaptation actions, adaptive capacities, and the overall role of
smaller local businesses in climate adaptation. In comparison, small and micro businesses typically
have lower profits, smaller cash reserves and seldom backup resources so that a single extreme
weather event led to long-lasting negative impacts. Clearly, smaller businesses lack the capacity to
design and implement adaptation measures (Zhang et al 2009).Small and micro businesses are
therefore bearing the brunt of climate-related impacts – a burden that is expected to intensify over
the next decades (e.g., Lo et al., 2019; Ngin et al., 2020). In the area of today's risk from flooding, storm
surges, and heavy rainfall, several studies illustrate that smaller businesses with local operations in
particular experience both direct impacts like property damage, mechanical breakdowns, and the
destruction of stocks and assets, as well as indirect impacts like postponed distribution and
interruptions of business operations and supplies (Bahinipati et al., 2017; Marks and Thomalla, 2017;
Neise et al., 2019; Verrest et al., 2020; Wedawatta et al., 2014; Wedawatta and Ingirige, 2012). In
addition, they are often situated in a multi-risk environment, usually unprotected by public flood
protection. This is especially true in HCMC where uncontrolled urban expansion since the beginning of
the 21$^{st}$ century into flood-prone areas led to increased exposition. Poorly established and connected
infrastructure has exacerbated flooding risks leading to a reduction in water regulation capacity,
drainage capacity, water permeability, and land subsidence (Storch and Downes, 2011; The World
Bank, 2019). As a result, small and micro businesses are forced to respond to climate risks
independently due to their higher vulnerabilities (Lo et al., 2019).
Recent research has sought to determine whether and to what extent micro businesses are responding
to acute climate risks such as flooding and what options they have to prepare for the intensification of
future impacts. Ngin et al. (2020) show that micro businesses in the tourism and hospitality sector in
Cambodia usually adopt temporary and reactive responses against floods and storms rather than long-
term and proactive strategies. In the same vein, Neise and Revilla Diez (2019) emphasize that most of
the small and micro manufacturing firms in their case study in Jakarta only cope during a flood event.





Relying on their established routines, they use floodwalls and sandbags to protect their production
facility from water, place their products in higher places, and use small pumping systems to drain the
water. While Chaudhury (2018) makes some arguments for motivating businesses to take proactive
adaptation measures (e.g., greater risk awareness, benefits of adapting outweigh the financial costs),
micro businesses face several barriers and structural deficits that limit their adaptive capacity and
decision to invest in individual adaptation measures. Unlike their larger counterparts, whose
adaptation actions are usually driven by organizational characteristics, such as financial liquidity,
business performance, foreign ownership and knowledge-spillovers, micro businesses are facing
different barriers (Leitold et al., 2021; Lo et al., 2019).

**2.2 Drivers and barriers of micro business adaptation**
Based on the initial findings in vulnerability and adaptation literature, we present a conceptual
framework to help to understand drivers and barriers to adaptation action of micro businesses. Many
micro businesses find it challenging to develop adaptation strategies because of four key barriers (see
Figure 1).

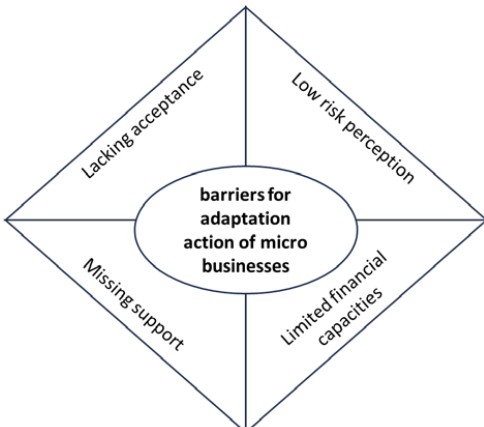


**Figure 1:** Key barriers for adaptation action of micro businesses
*Key barrier 1: Acceptance of adaptation measures*
A key barrier to addressing climate risks is lacking acceptance of adaptation options due to cultural
attitudes, social barriers, and a lack of understanding. A study by Geaves and Penning-Rowsell (2016)
shows that large-scale protection measures fail to attract long-term participation from private actors
due to a lack of local bonding. By contrast, a recent study by Leitold et al. (2020) reveals that small-
scale adaptation measures initiated in a smaller neighborhood, like flood protection awareness
programs, can promote the willingness of resident SMEs to adapt. In particular, collaborative
approaches, with shared funding by different actors (i.e., the community, firms in the neighborhood,
local government gives incentives) could help to overcome biases, and support the implementation of
different adaptation options. Understanding an adaptation measure, its tangible costs, and benefits
can lower the social barriers to adopting new technologies and participating in flood adaptation
(Chaudhury, 2018).
*Key barrier 2: Risk perceptions at individual and household level*
Since micro businesses are "owner-centered" (Gherhes et al., 2016), individual risk perceptions, skills
and capabilities, and experience with, for example flood impacts, of decision-makers play an important





role in micro business adaptation. Lawrence et al. (2014) reveal that flood experiences at the individual
household level in New Zealand contribute to increased risk perception and readiness to adapt. For
the manufacturing sector, Neise et al. (2019) also show that SME adaptation to flood impacts in
Indonesia is closely related to the risk prepardness of their managers. However, information on future
climate impacts are often inaccessible for micro businesses or even completely absent at the local
level, leading businesses to make decisions based on subjective perceptions (Chaudhury, 2018;
Danielson and Scott, 2006). In addition, there is general agreement that individual and household
education levels can influence how businesses are organized and managed, and how they respond to
current and future climate risks (Chirico and Salvato, 2008; Crick et al., 2018; Lo et al., 2019). Yet, the
link between business viability and the need to adapt to future climate change impacts is not suffiently
visible (Frei-Oldenburg et al., 2018).
*Key barrier 3: Financial capacities*
It is common knowledge that business characteristics are critical factors that shape adaptive action
(Agrawala et al., 2011; Halkos et al., 2018; Pulver and Benney, 2013). In particular, limited financial
resources and business performance have been proven to be barriers to the implementation of
adaptation measures in small and micro businesses. They tend to have lower business capital and cash
reserves, and are less likely to have financial reserve funds (Gherhes et al., 2016). A study by Marks
and Thomalla (2017) shows that after severe flooding in Thailand in 2011, SMEs recovery was
particularly hampered by financial constraints. Chaudhury (2018) further argues that even after
conducting robust risk assessments and identifying cost-effective adaptation options, limited financial
capacity hinders the actual implementation of planned measures. In addition, the direct business
neighborhood could shape collective business adaptation as micro businesses are highly dependent
on their local customers and suppliers. Leitold et al. (2020) illustrate that interaction with neighboring
firms is a driving factor for SMEs to invest into collective local adaptation measures. In the same vein,
Pauw and Chan (2018) argue that smaller businesses could take active responsibilities in localized
initiatives that connect different actors in the same neighborhood.
*Key barrier 4: Influence of the institutional environment*
Obviously, there are broader structural deficiencies in external support for microbusiness adaptation
financing. In most recent literature that is developing around disaster risk reduction and adaptation
barriers, access to and use of external finance such as loans and credits from banks or microcredit
institutions and tailored insurances is argued to be the major bottleneck for adaptation (Chaudhury,
2018; Chinh et al., 2016; Crick et al., 2018; Surminski and Hankinson, 2018; UNDP, 2019; UNDRR, 2020).
As many micro businesses are part of the informal economy, social protection and external financing
mechanisms are often not accessible at the business level (UNDRR, 2020). Therefore, it is not surprising
that Halkos et al. (2018), Neise et al. (2019), and Leitold et al. (2020) found that institutional support
and external guidance have a direct impact on the engagement of smaller firms in implementing
adaptation measures against recent and future extreme events like floods and storms. In some
economies like Vietnam, private businesses are underserved with respect to supportive policies and
regulations (Revilla Diez, 2016; Trinh and Thanh, 2017). Therefore, local (business) associations have
recently been considered as a promising information channel around climate change impacts and
ultimately for stimulating adaptation action of private businesses.



**3    Material and methods**
**3.1 Study area: Flooding in HCMC and the impact on micro businesses**
HCMC in Vietnam is already experiencing frequent flooding, which is expected to intensify in the
coming years and decades due to the impacts of climate change. Seasonal extreme rainfall, storm
surges, and discharge from upstream reservoirs often come at the same time with high tides and
rainfall peaks, already resulting in compound flood events in many parts of the city (Downes and
Storch, 2014; Scussolini et al., 2017). Located on the north-eastern edge of the Mekong Delta, at the
mouth of the Dong Nai river basin, HCMC is characterized by topological conditions like many other
delta regions in the world. More than half of the city is situated lower than 1.5 meter elevation above
mean sea level (ADB, 2010). Lowlying lands, proximity to the sea, and an interconnected system of
small rivers and chanels result in a high overall exposure to future sea-level rise. According to national
studies, the sea level has already risen by 20 cm off the coast of Vietnam in the last 50 years before
2009 (MONRE, 2009) and the trend is upward (Scussolini et al., 2017). In addition, uncontrolled urban
expansion and poorly connected infrastructure act as flood risk multipliers, leading to land subsidence,
and a reduction in drainage capacity and water permeability. This is particularly problematic during
the rainy season (May to October), which already provides 85 % of the total rainfall per year (MONRE
et al., 2006; World Bank, 2019).

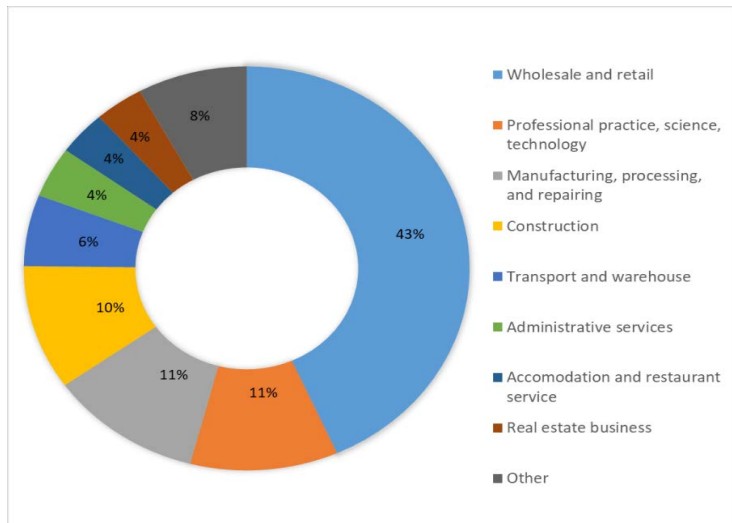

**Figure 2:** Main sectors of micro businesses in HCMC [percentage of businesses]
Source: GSO (2017)

Set in the motion by the liberalization policies in 1986 and the subsequent transition to a market-
oriented economy, HCMC has been steadily growing in population and private businesses. As the
Vietnam's largest city, HCMC is home to officially 8.9 million people (GSO, 2020). Although the private
economic sector plays a decisive role in HCMC's remarkable economic development, many of the SMEs
and micro businesses are at the forefront of flood-related losses and damage. In HCMC, 43 % of the
approximately 145,000 officially registered micro businesses (1-9 employees) in 2017 belong to
wholesale and retail, 11 % to manufacturing, processing, and repairing, and 10 % to the construction
sector (GSO, 2017, see Figure 2). Micro and family businesses in these sectors, in particular, are highly
exposed to recent and future flood impacts. Manufacturing businesses are sensitive to compound



flooding sources due to their location-specific production, hard-to-change infrastructure, and heavy
machinery. While many medium and large firms operating in international value chains are often
located in industrial parks with sufficient infrastructural flood protection, small and micro businesses
have to put up with business interruptions during flood events several times a year (Leitold et al.,
2021). Wholesale and retail businesses are highly dependent on regional and local value chains, which
are particularly disrupted by heavy rainfall during the rainy season and tide-induced flooding. In
addition, direct flooding in stores damages flood-sensitive goods such as flowers, food, and paper
products.
This study was carried out in four case study areas in HCMC where households and micro businesses
have already suffered some flood damages in recent years. Businesses in the western part of the city
(District 8 and Binh Tan) are mainly impacted by urban flash floods and pluvial flooding after heavy and
prolonged rainfall. Binh Thanh district is located close to the Sai Gon river, making the area exposed to
flooding, which is exacerbated by the release of upstream water reservoirs (Duy et al., 2018). Nha Be
district is located in the southern parts of the city and characterized by a peri-urban morphology.
According to the projections, Nha Be is one of the districts in HCMC that will be most affected by future
sea level rise (Scussolini et al., 2017).

### 269    3.2 Household survey and scenario-based field experiments

The empirical analyses in this paper are based on two combined datasets.
First, we used a household survey in HCMC conducted as part of a collaborative research project
"DECIDER" (*Decisions for the Design of Adaptation Pathways and the Integrative Development,*
*Evaluation, and Governance of Flood Risk Reduction Strategies in Changing Urban-Rural Systems*) to
understand flood vulnerability of micro businesses, their respective perceptions, and flood adaptation.
The standardized household survey was conducted in two different wards of the four case study
districts (District 8, Binh Tan, Binh Thanh, and Nha Be) in HCMC. In addition to 748 households, a total
of 252 micro businesses were surveyed in HCMC between September and November 2020. We
developed a questionnaire on general characteristics and the economic situation of micro-businesses,
investment decisions, flood impacts, adaptation strategies and perceptions of future risk und local risk
management systems. All respondents have been experienced with flooding (i.e., water entering the
house/business premise) and suffered damages/losses due to floods since 2010. The questionnaires
were field tested during a one-week pretest in 2019, and adjusted afterwards. Moreover, the survey
was preceded by a one-day workshop for the enumerators during which they were trained how to
conduct the survey and received feedback. In Vietnam, our partners of the Southern Institute of Social
Sciences (SISS) organized and implemented the training and the main field campaign.
Second, we run scenario-based field experiments with about a quarter of micro businesses owners
from the main survey. The goal of the experiments was to examine the willingness of businesses to
invest in collective adaptation options to protect themselves from future flood impacts. The scenario-
based field experiments consist of a public-good game design with different adaptation scenarios in a
field-experiment environment (Leitold et al., 2020; Neise et al., 2019). Public-good games are rooted
in behavioral economics. They aim to explain why collective actions succeed or fail and decipher
participants' contributions to a public good (Ones and Putterman, 2007). In our experiments, flood
adaptation measures are defined as discrete public goods that are only provided when multiple actors
make individual financial contributions. Implementing public good games in real field environments
rather than in a laboratory, provides a deep understanding of explanatory factors for participants'
decision making in collective adaptation actions (Ehmke and Shogren, 2009). The experiments used



vignette designs that present carefully constructed but hypothetical descriptions of adaptation
measures that differ in their design and the financial contributions for their implementation (Atzmüller
and Steiner, 2010).
In total, our Vietnamese project partners from the University of Economics and Law, Vietnam National
University conducted experiments with 62 out of the 252 micro businesses from the main survey. The
methodology, and the different scenarios were explained in detail to the enumerators in a training
workshop and during supervised pre-tests prior to the experiments. Then, we linked the micro business
survey data to our experiment data using the survey identification to combine information on
household and business characteristics and perceptions with the investment decision at each
experiment (see Figure 3).

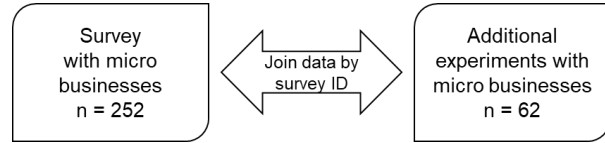

**Figure 3:** Data basis for the multilevel regression analysis

**3.3 Experiment design, measurement, and data analysis**
Four realistic adaptation measures were developed for the experiments. The conceptualization of
these measures is based on our previous study of manufacturing SMEs in HCMC (2018-2019, see
Leitold et al., 2020), but was adapted to the local realities of micro businesses in collaboration with our
local project partners.
To analyze the influence of respective adaptation measures and financing options on the willingness
of micro businesses to participate in collective adaptation, we used the same locational setting
representing typical-flood prone areas in HCMC for all adaptation options (see **Figure 4** for the overall
experiment setting). We designed four adaptation cards, which were shown to the participants. In
addition to the designs of adaptation measures, we built different financing options into the scenarios
cards. In the first two options, either the residents in the neighborhood or the other micro businesses
contribute to the same amount and share the costs of adaptation measures *(neighborhood support)*.
In the third option, local authorities provide financial incentives and support the implementation of
adaptation measures *(political support)*. By contrast, in the fourth option, local authorities demand the
participation of businesses or impose fines for non-compliance *(political pressure)*. In the fifth option,
other businesses contribute less than the necessary amount and the micro business must invest more
than others in their direct neighborhood *(unbalanced contribution)*. In total, the respondents have to
go through 20 scenarios (4 adaptation scenarios multiplied by 5 financing options).
For data analysis, we created a dichotomous dependent variable *willingness to participate in collective*
*flood adaptation*, where '1' was coded for micro business is willing to invest the necessary resources
and '0' that a micro business was not willing to contribute sufficiently. In general, our indicators are
presented on a binary scale (see Table 1 for the explanation of indicators). Following Leitold et al.
(2020), we tested for *dike systems, drainage systems*, and *awareness programs* to assess the
acceptance of different adaptation measures **(key barrier 1: lacking acceptance)**. To test preference
for different funding options, we used *neighborhood support* as a proxy for the preference for shared



funding of measures, and *political support* as a proxy for desired support from public stakeholders. We
also controlled for *unbalanced contributions of businesses.*

**Adaptation Option 1: Dike construction**    **Adaptation Option 2: Drainage system**

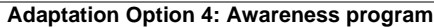

**Measure:**
Sophisticated dike system with two sluices in front of the river. Retention area in front of the riverside

**Strategy:**
Safeguard riparian zones and canal areas. Reduce inundation level

**Measure:**
Clean and upgrade drainage system within the community

**Strategy:**
Increase drainage capacity. Reduce inundation level

**Adaptation Option 3: Road elevation**    **Adaptation Option 4: Awareness program**

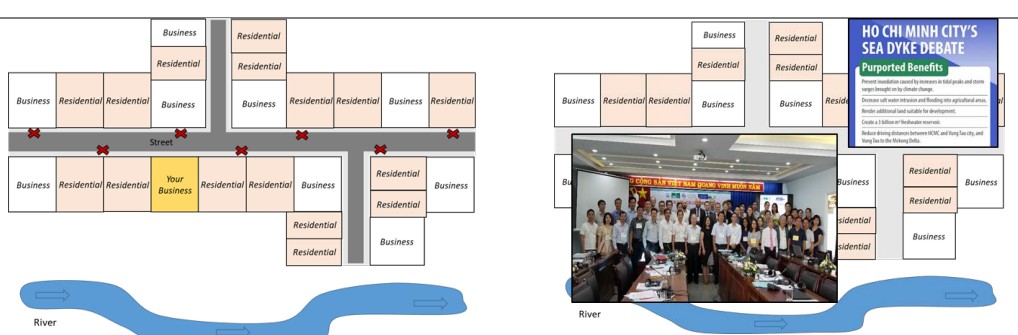

**Measure:**
Elevation of the main roads in the neighborhood

**Strategy:**
Reduce inundation level due to heavy rain.

**Measure:**
Funding of an awareness raising program
Develop district adaptation pathways

**Strategy:**
Strengthen flood risk management. Increase awareness on flooding (and waste disposal etc.)

**Figure 4:** Overall experiment setting and adaptation options (Source: the design is based on Neise et
al., 2019; and Leitold et al., 2020)
To test for the influence of risk perceptions at individual and household level **(key barrier 2: low risk**
**perception)**, we generated the indicators *expected flood increase* and *household education* (Crick et
al., 2018). The latter describes that at least one person of the household has a university degree or
vocational training. Consistent with Neise et al. (2019) and based on the assumptions of Lawrence et
al. (2014), *flood experience* was measured whether a micro-business was flooded more than five times
in the last 10 years. Based on the answers from the micro-business survey, an additional measure of
future flood perception was included to represent *high individual damages* that occurred during the





most serious flood since 2010. We hypothesized positive relationships between the indicators for risk
perception at individual level and the willingness to participate in collective adaptation.
In the business environment, we tested for financial capacities as factor influencing adaptation
decisions **(key barrier 3: limited financial capabilities).** Following Chaudhury (2018) and Marks and
Thomalla (2017), we developed indicators of *decline in business revenue* (when revenues have declined
or fluctuated over the past five years) and *limited financial resources* (self-assessment of micro
businesses of their financial resources for flood adaptation). We expect both indicators to be barriers
to collective adaptation. We also tested dependence on local customers and suppliers as relations with
neighboring firms raises the probability for co-funding by other firms, although this indicator is difficult
to operationalize. However, we coded *local supplier* as '1' for businesses that report that their suppliers
are located in the same flood exposed neighborhood.

**Table 1:** Key indicators of collective flood adaptation

| | Indicators | Descriptions (No=0; Yes =1) | Expected impacts |
|---|---|---|---|
| Adaptation measure | Neighborhood support | Scenarios with shared funding | + |
| | Political support | Scenarios with shared funding | + |
| | Unbalanced contribution of businesses | Scenarios where micro businesses need to invest more than others in their neighborhood | - |
| | Dike system | Scenarios with dike system (high financial input, technological infrastructure) | - |
| | Drainage system | Scenarios with drainage system (medium financial input, technological infrastructure) | + |
| | Awareness program | Scenarios with awareness program (low financial input, soft measure) | + |
| Individual / Household | High individual damages | High damage of business components (e.g., furniture, electronics, equipments, products), high equals major and complete damage | + |
| | Flood experience | Business was flooded more than 5 times in the last 10 years | + |
| | Household education | At least one person of the household has a university degree or vocational training | + |
| | Expected flood increase | Expected flood incease in the next ten years | + |
| Business environment | Decline business revenue | Revenue decline/fluctuation in the last five years | - |
| | Limited financial resources | Low financial resources for preventing flood impacts (rating from 1-5, low equals 1 and 2) | - |
| | Local supplier | Suppliers located in the same district | + |
| Institutional environment | Member organization | Household members are part of an organization (e.g, Fatherland's Front, Women's Union, Youth Union, etc.) | + |
| | No repair after flood events | Government/Law doesn't allow to repair/rebuild after flood events (e.g. it is in a planning project area) | + |
| | Access to external capital | Business finances investments through loans from banks or microcredit institutions | + |


We test the influence of the institutional environment using three explanatory variables **(key barrier**
**4: missing support).** It is expected that the willingness to participate in adaptation if a *household*
*member is part of an organization* (i.e., Fatherland's Front, Women's Union, Youth Union) (Leitold et



al., 2020). Especially, in Vietnam being a member the party's own social organization could offer some
patronage and special treatment when it comes to applying for support. To represent institutional
barriers, we build an indicator for the situation where *public policies or public laws do not allow private*
*buildings to be repaired or rebuilt* after floods. For example, when micro businesses are located in a
planning project area which is quite common in HCMC in recent years. Further, we test the influence
of *access to external capital* in the form of loans from banks or microcredit institutions on willingness
to participate. We expect negative correlations for both indicators and adaptation willingness. Finally,
we controlled for the influence of location within our four case study areas.
The scenario-based field experiments generated 1,240 observations for data processing. As each
participant responded to 20 scenarios, scenario data are nested within business characteristics.
Analyzing such hierarchically structured data with ordinary least squared regression would lead to
spatial autocorrelation and a violation of the independence assumption for scenario observations (Hox
et al., 2017; Sohns and Revilla Diez, 2018). Therefore, we applied a two-level binary-logistic regression
that allows us to consider the differences and interdependencies between scenario and micro business
characteristics (Rabe-Hesketh and Skrondal, 2008). Multicollinearity (average variance inflation factor
for the independent variables: 1.6) can be rejected.

**4 Findings**
**4.1 Descriptive findings**
Our sample consists of 62 micro businesses. 46 businesses are stores or retailers (74 %) for food and
beverages, clothing, houseware, electricity, or construction material. 10 businesses are operating in
the service sector (16 %), and three in the production sector (5 %). 61 % of all businesses have been
flooded more than five times per year in the last 10 years, and 44 % even more than 10 times a year.
It is evident that as soon as damage is reported, it is mostly classified as major damage requiring repair.
In particular, the level of damage to products is relatively high (see Table 2). Complete damage has not
been reported. As a consequence, the micro-businesses do undertake own precautionary measures.
We see that more than 50 % of the micro businesses already purchased water barriers for flood
prevention and dry-proof their valuables, goods, and products during severe flood events. In addition,
84 % of micro businesses have already elevated their ground floor or foundation to prevent flooding
into their premises. In terms of acute flooding events, which are already clearly noticeable today, the
micro businesses are therefore (most reactively) already doing something.
**Table 2**: Individual damage of micro businesses from the most severe flood since 2010

| | no damage | minor damages | moderate damages | major damages-needs repair | complete damage - needs replacement | no answer |
|---|---|---|---|---|---|---|
| Furniture | 39 | 8 | 1 | 14 | 0 | 0 |
| Electronics | 37 | 3 | 4 | 16 | 0 | 2 |
| Business specific equipment | 39 | 6 | 8 | 9 | 0 | 0 |
| Products | 28 | 4 | 6 | 22 | 0 | 2 |


The descriptive analysis of the key barriers partly confirms, but also oppose the findings from the
literature.
In respect to the key barrier 1 (lacking acceptance), the complete rejection of adaptation measures
cannot be confirmed as stated in the conceptual section. However, in only 28 % of all scenarios, micro



businesses were willing to contribute to flood adaptation measures in their neighborhood. The results show substantial differences between participation in technical scenarios (dike system: 29 %, drainage system: 25 %, elevation: 26 %) and the less expensive flood awareness program, to which micro business owners were willing to contribute in 68 % of cases (see Figure 5). In terms of financing adaptation measures, decision-makers were willing to contribute financially in 39 % of the scenarios if other actors in the neighborhood (i.e., the community: 30 % and other businesses: 48 %) were also involved. For all other options - financial support from local authorities or when businesses have to pay a fine for not investing in collective protection measures - willingness to participate was below average (see Figure 6). Also, the results for the key barrier 2 (low risk perception) are different than expected. The survey results indicate that 77 % of the businesses expect flooding to increase in the next 10 years, while 16 % expect flooding to remain the same or even decrease. These results underline that owner of micro-businesses are well aware of the risks of future flooding.

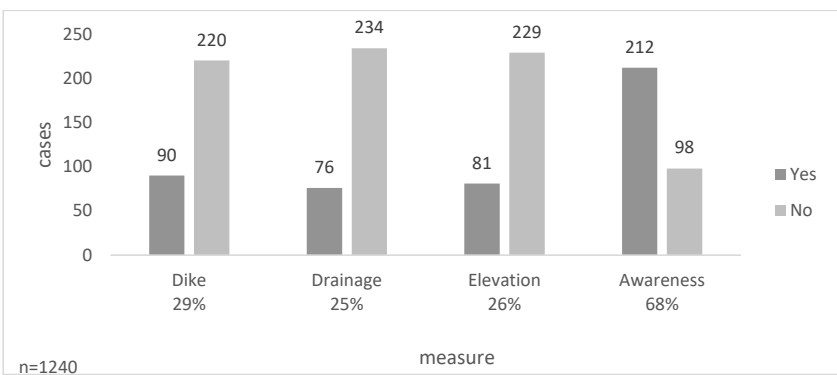

**Figure 5:** Preference of adaptation measure

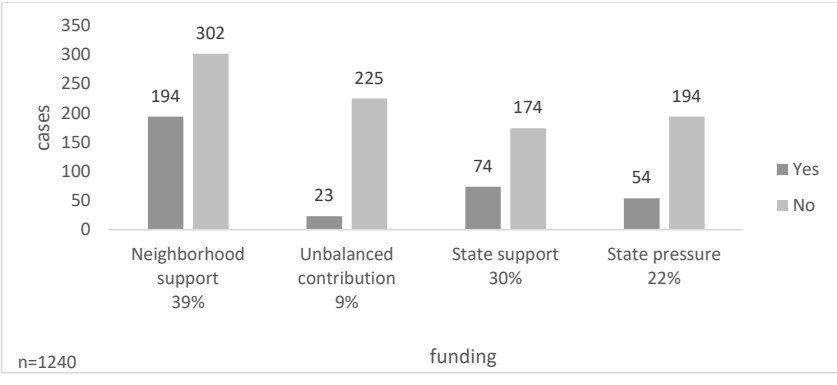

**Figure 6:** Preference of funding

In relation to the key barriers 3 (limited financial capabilities) and 4 (missing support) the results are in line with the findings in the literature. About 37 % of businesses report a decline, and 15 % fluctuation in business revenues over the past five years. In addition to revenue, the financial resources available to prevent flood impacts are a key limiting adaptation factor to micro businesses. On a scale from 1 (very poor) to 5 (very good), more than half of the businesses rate their financial resources as limited (58 % rate 1 and 2). Only 19 % of businesses surveyed have access to external capital, such as loans



from banks or from microcredit institutions. 16 % receive loans from family members, relatives, or
friends, while the majority finance their business investments through personal funds or savings.
Support by state agencies is hardly mentioned.


**4.2 Multilevel regression findings**
In order to detect the key drivers and barriers for micro-business adaptations'` strategies, the main
statistical analysis was based on the two-level regression. **Table 3** shows which indicators influence the
willingness of micro businesses to invest in collective flood adaptation measures and whether they act
as either drivers or barriers to adaptation. The scenario-level results underscore the findings of the
descriptive analysis. Micro businesses significantly prefer to invest in the awareness program, while
their willingness to invest is not influenced by hard technical measures like, for example, the dike
system or the drainage system. What is particularly clear is that shared funding opportunities between
micro businesses and local authorities, as well as the community and other businesses in their
neighborhood, significantly increase the investment in collective flood adaptation. Accordingly, an
unbalanced contribution of businesses in their neighborhood reduces the investment and thus acts as
a barrier.
In addition, further variables also influence the willingness to participate in adaptation measures.
Businesses that already suffered high individual damages during the most serious flood since 2010,
businesses that have high flood experience, and those that expect a high increase in floods in the next
ten years are significantly more willing to invest in collective flood adaptation measures. Since the
influence of high individual damage yields positive, but only slightly significant results it should be
interpreted with caution. In the overall picture, all three indicators of risk perception and experience
act as drivers for collective adaptation. Interestingly, the investment decisions of micro businesses
were not influenced by household education.
As expected, financial constraints and decreasing business performance indicators act as barriers for
collective adaptation. A general decline in business revenues and limited financial resources for
adaptation measures are reflecting the situation of the majority of businesses in the sample. Both
indicators significantly decrease participation in the scenarios. Regarding the dependence on local
suppliers, the analysis did not yield any significant results.
The results further reveal that external guidance and institutional support play a major role in micro
business decision making for collective adaptation. When a household member is part of an
organization, the willingness to invest in collective adaptation increases slightly significantly. Similarly,
access to external capital in form of loans from banks or microcredit institutions increases the
willingness to participate. Since some urban development policies act as barriers to individual risk
reduction and hinder the repair or reconstruction of private buildings after flood events, it is not
surprising that such situations have highly significant positive influence on the willingness to invest in
collective adaptation measures, together with other actors in the neighborhood.
The neighborhood of micro-businesses on decision-making in the experiments for which we controlled
does not yield significant results. Thus, micro businesses in the case study areas make decisions based
on scenario and individual-level characteristics, regardless of their place of operation.




**Table 3:** Multilevel regression results for willingness to participate in collective flood adaptation

| Fixed effects | Odds ratio (standard error) m0 | Odds ratio (standard error) m1 | Odds ratio (standard error) m2 | Direction of effect |
|---|---|---|---|---|
| *Scenario characteristics* | | | | |
| Neighborhood support (shared funding) | | 4.721*** (1.207) | **4.712*** (1.208)** | + |
| Political support (shared funding) | | 2.222*** (0.638) | **2.231*** (0.643)** | + |
| Unbalanced contribution of businesses | | 0.133*** (0.055) | **0.121*** (0.052)** | - |
| Dike system | | 1.333 (0.338) | **1.334 (0.338)** | |
| Drainage system | | 0.845 (0.220) | **0.844 (0.220)** | |
| Awareness program | | 1.697** (0.039) | **1.697** (0.426)** | + |
| | | | | |
| *Firm characteristics* | | | | |
| High individual damages | | | **3.207* (1.964)** | + |
| Flood experience | | | **5.596** (4.158)** | + |
| Expected flood increase | | | **7.496** (6.541)** | + |
| Household education | | | **1.322 (0.808)** | |
| Declining business revenue | | | **0.167** (0.121)** | - |
| Limited financial resources | | | **0.189** (0.126)** | - |
| Local supplier | | | **2.523 (1.759)** | |
| Member organization | | | **4.673* (4.184)** | + |
| No repair after flood events | | | **193.237*** (252.860)** | + |
| Access to external capital | | | **4.394* (3.624)** | + |
| | | | | |
| *Control variables* | | | | |
| Nha Be (location) | | | **3.136 (2.750)** | |
| District 8 (location) | | | **2.239 (1.930)** | |
| | | | | |
| Constant | -1.489 (0.281) | 0.894 (0.386) | **0.000*** (0.000)** | |
| | | | | |
| Random effects | | | | |
| Firm var.(_cons) | 4.364 (1.146) | 6.938 (1.840) | **3.780 (1.020)** | |
| | | | | |
| Model fit statistics | | | | |
| Observations | 1,240 | 1,240 | **1,240** | |
| ICC firm | 0.570 | 0.678 | **0.535** | |
| Prob>chi2 | 0.000 | 0.000 | **0.000** | |



***Significant at 1% level (p<0.01); **Significant at 5% level (p<0.05); *Significant at 10% level (p<0.1). Source:
Own calculation

## 5     Future role of micro businesses in collective flood adaptation

The empirical results of this analysis add important insights from the particular case of HCMC toward
a broader understanding of drivers and barriers of micro business flood adaptation.
The acceptance of and participation in adaptation measures are clearly related to the risk perceptions
and awareness at the individual and household level. In this case study, high future risk perception,
often based on past experience with flooding and suffering from damage to stocks and assets, was
clearly identified as a driver for investment in collaborative flood adaptation. Conversely, a lack of risk
perception and assessment, particularly with an eye towards upcoming flood risks, acts as a barrier for
long-term adaptation. Although 77 % of the businesses in our survey expect flooding to increase –
suggesting that the awareness is quite high – the direct (or indirect) impact on business operations is
often unclear and may explain the overall restraint in the experiments. Schaer (2018) argues that either
businesses do not perceive future impacts to be a risk factor for their business operations or have
limited expertise to predict and plan the risks accurately. The link between business benefits and
adaptation is not clear to decision-makers. It is added that micro businesses differ from larger SMEs by
being owner-centered, having a tendency of being "growth-adverse", and focusing more on non-
economic aspects of business ownership. Growth intensions are often limited to a desired income
which is sufficient for making a living (Gherhes et al., 2016). Neise and Revilla Diez (2019) and Leitold
et al. (2021) already point out that frequent but smaller floods are kind of normality for small
businesses, against which they do not plan to adapt in the future. They often lack long-term business
plans or any risk assessments, either for climate risks or for other business risks, and follow a "simply
live with it" attitude. Business growth in terms of increasing headcount, diversification of products and
services, and revenue growth is not aspired anyway. Thus, the impact of flooding is only relevant if it
threatens the profitability of the micro business for household income.
Following this vein, we clearly see an overlap of the different key barriers developed in our conceptual
framework (Lo et al., 2019). It can be argued that general development constraints of micro businesses
are also responsible for barriers to adaptation. In particular, financial limitations in the business
environment act as additional barriers for long-term adaptation. On this point, the institutional
environment represents another critical barrier that can stimulate or inhibit adaptation. There is a lack
of tailored external support mechanisms and adequate financing options that motivate micro
businesses to initiate long-term business planning and thus also enables them to implement
adaptation measures (Berkhout et al., 2006; Schaer, 2018).
In general, the willingness to participate financially in our scenario exercise stood at 28% and was lower
than what we had expected. Average results in such public good games typically amounts to 40-60 %
of personal endowment (Chaudhuri, 2011). The results of the experiments show no substantial
differences between the contribution to different technical adaptation measures and the influence on
decision-making to participate in adaptation measures. Although the preference for low-cost and soft
measures over cost-extensive and technological measures is generally comparable to experiments
with manufacuturing SMEs (Leitold et al., 2020; Neise et al., 2019), the low uptake of technical
adaptation measures can be explained by micro businesses' prerequisites like limited financial
capabilities and low risk perception for entrepreneurial decision-making.
However, depending on the adaptation measure and financing option micro-businesses could play a
larger role in flood adaptation. Overall, almost 70% of the micro-businesses are willing to participate
in collective awareness programs. In general, the willingness to participate financially increases



noticeable to 39% if the costs could be shared with actors in their neighborhood and local authorities.
Moreover, businesses that have access to external capital from banks or microcredit institutions are
more willing to participate in collective adaptation in general. In most cases, and in contrast to larger
firms, micro businesses have a local life and business horizon and are closely embedded in local
(business) networks (Halkos and Skouloudis, 2019; Kato and Charoenrat, 2018). Therefore, local
adaptation solutions, support mechanisms and incentives must also be created in the direct business
environment. Building local business associations outside of industry-specific associations and
engaging decision-makers could be one important starting point to involve micro businesses into larger
adaptation initiatives and motivate them to participate. Additionally, community organizations and
neighborhood unions should place future risk trends and flood hazards on their agendas to promote
micro business awareness of flooding, but also support micro businesses that face institutional barriers
for flood adaptation.
It is argued here and supported by Chaudhury (2018), that information about future climate-related
risks and uncertainties, while relevant for decision-making processes, is often still unavailable for micro
businesses. Therefore, additional initiatives like awareness raising programs are easy to implement
and do not reach technological capacity limits, but can help to promote future risk assessments and
weighting of adaptation options, and possibilities. Building effective adaptation infrastructure consists
not only of physical infrastructure such as elevation, drainage systems or dike systems, but needs also
to include "informational infrastructure" (Marlowe et al., 2018; Ngin et al., 2020) in the form of
channels for communicating disaster risks and raising awareness. But apparently as our result clearly
show, micro-businesses willingness to participate in adaptation is also subject to socio-economic
constraints confronting individual decision makers and their lifestyle preferences (Lo et al 2019).
This understanding of micro-businesses, their lifestyle orientation and their flexibility is often
overlooked in adaptation research and in adaptation policies (Parsons et al 2018). There is a need to
understand more about constraints and preferences of micro-businesses to better support them but
also to integrate them better in adaptation schemes. As they are often located in densely populated
neighborhoods where they also reside and form part of the social fabric, their role as multiplier for
collective action could be used strategically in adaptation plans. However, our analysis is just a first
step into this direction. Our multilevel analysis is based on hypothetical and simplified designs of
adaptation scenarios. Therefore, external validity should be improved by conducting similar
experiments in different field contexts. Moreover, the research design based on yes or no responses
is not able to capture the intensity of contextual influences on micro business' willingness to participate
in respective adaptation options. Another relevant future research avenue is to quantitatively
investigate the causal relationships of various drivers and barriers that influence micro business
decision-making for flood adaptation based on a higher number of experiments.

**6    Conclusion**
Micro-businesses could play a much larger role in collective adaptation. Often overlooked in
adaptation research, their willingness to contribute in collective action amidst major constraints is
surprising. The conceptual framework presented in this paper helps us to understand the key drivers
and barriers of micro-businesses willingness to participate in collective adaption activities. The most
important key barriers of micro-businesses are limited financial capabilities and missing support from
local authorities. However, micro-businesses are willing to contribute depending on the concrete
adaptation measure and funding options. If no financial contribution is expected, almost 70 % are
willing to assist in awareness raising campaigns. And although their financial capabilities are very
limited, 39% of the micro-businesses would contribute financially if the costs are shared with other



firms in their neighborhood and with local authorities. Against this background, micro-businesses
should be much more involved in adaptation plans and measures. Through their local embedding, they
can be important multipliers in strengthening adaptive capacity at the local level.



**Appendix:**

| | Indicator | Description (min=0; max=1) | Obs. | Mean | Std. Dev. |
|---|---|---|---|---|---|
| **Adaptation measures** | Neighborhood support | Scenarios with shared funding | 1,240 | 0.4 | 0.49 |
| | Political support | Scenarios with shared funding | 1,240 | 0.2 | 0.40 |
| | Unbalanced contribution of businesses | Scenarios where micro businesses need to invest more than others in their neighborhood | 1,240 | 0.2 | 0.40 |
| | Dike system | Scenarios with dike system (high financial input, technological infrastructure) | 1,240 | 0.25 | 0.43 |
| | Drainage system | Scenarios with drainage system (medium financial input, technological infrastructure) | 1,240 | 0.25 | 0.43 |
| | Awareness program | Scenarios with awareness program (low financial input, soft measure) | 1,240 | 0.25 | 0.43 |
| **Individual risk knowledge, risk assessment and flood experience** | High individual damages | High damage of business components (e.g., furniture, electronics, equipments, products), high equals major and complete damage | 1,240 | 0.52 | 0.50 |
| | Flood experience | Business was flooded more than 5 times in the last 10 years | 1,240 | 0.61 | 0.49 |
| | Household education | At least one person of the household has a university degree or vocational training | 1,240 | 0.52 | 0.50 |
| | Expected flood increase | Expected flood incease in the next ten years | 1,240 | 0.77 | 0.42 |
| **Business environment** | Decline business revenue | Revenue decline/ fluctuation over the past five years | 1,240 | 0.51 | 0.50 |
| | Limited financial resources | Low financial resources for preventing flood impacts (rating from 1-5, low equals 1 and 2) | 1,240 | 0.58 | 0.49 |
| | Local supplier | Suppliers located in the same district | 1,240 | 0.60 | 0.49 |
| **Institutional environment** | Member organization | Household members are part of an organization (e.g, Fatherland's Front, Women's Union, Youth Union, etc.) | 1,240 | 0.15 | 0.35 |
| | No repair after flood events | Government/Law doesn't allow to repair/rebuild after flood events (e.g. it is in a planning project area) | 1,240 | 0.07 | 0.25 |
| | Access to external capital | Business finances investments through loans from banks or microcredit institutions | 1,240 | 0.18 | 0.38 |
| *Control variables* | Spatial influence Nha Be | Business located in Nha Be | 1,240 | 0.44 | 0.50 |
| | Spatial influence District 8 | Business located in District 8 | 1,240 | 0.32 | 0.47 |




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
