# Peer review of "Micro business participation in collective flood adaptation. Lessons from scenario-2 based analysis in Ho Chi Minh City, Vietnam"

_EGUsphere, 2023_

## Author Response (AR1)

Dear Paolo,

Many thanks for handling our paper. According to the comments by the reviewers, we tackled all the points raides by the reviewers. In the manuscript with track changes you can see all the changes we made.

With very best wishes

Javier

Reviewer 1:

The authors have presented survey results from micro businesses in HCMC in Vietnam regarding their willingness to participate and contribute toward combined flood adaptation and mitigation measures. The survey results are presented clearly and the authors' conclusions match those in previous similar research cited by them. I have only minor comments and few spelling errors for further improving the manuscript:

We express our gratitude to the Editor and the reviewers for conducting a thorough and positive review of our initial manuscript.

The authors have made a distinction between micro businesses and SMEs in the first sentence of their Introduction. However, later paragraphs apply the conclusions drawn by other authors for SMEs to micro businesses. A brief description of the distinction between micro business and SMEs, along with a short explanation of similarities that enable drawing the same conclusions would eliminate any confusion for the reader.

The authors have highlighted the importance of micro and small businesses by referring to them as building up the "social and economic fabric". A brief quantifiable description of their importance, such as, the percentage of people employed in them or percent of GDP that they contribute in HCMC, will help bolster the author's statement.

We express our gratitude to you for conducting a thorough and positive review of our initial manuscript. Your comment and observations will definitely improve the paper's coherence. Of course, we plan to take them up in our revisions.

**Response in respect to the differences between micro-businesses and SMEs**: As micro businesses are a specific subset of small and medium-sized enterprises (SMEs), microe businesses have both similarities and differences with larger small and medium-sized enterprises. Both micro businesses and SMEs are characterized by their relatively smaller size compared to larger firms, are typically privately owned and operated by entrepreneurs or a small group of individuals, and have a local or regional focus, serving a specific market or community. However, the literature suggests that micro businesses, by definition, are even smaller in terms of the number of employees, have lower sales and profits, and have limited assets. A systematic literature review by Gheres et al. (2016) shows that micro businesses often lack growth ambitions because owners tend to be growth averse and are constrained by underdeveloped skills in key business areas such as networking, marketing, business planning, and human resources. Due to time constraints, micro businesses are locked into day-to-day operations rather than investing time in long-term strategic business management. In addition, institutional bottlenecks place an additional burden on micro-enterprises. As a result, they have

limited access to higher-skilled labor, face a "closed" business environment as a result of negative external perceptions stemming from the stigmatization of their location, and find it more difficult to access finance and other support mechanisms than larger small and medium-sized enterprises.

In order to emphasize the uniqueness of micro-enterprises in the revision, we will try to make the distinction between micro-enterprises and larger SMEs clearer and be more cautious when referring to general SME studies. This explanatory paragraph has been added.

**Response to your comment on the importance of micro-businesses**: In the revision we will take this up. Here, the suggested statistics: The VN Census in 2020 shows that micro and small businesses still play an important role in Ho Chi Minh City. Alone, 86 % of the firms are micro-businesses. Small and medium sized businesses account for another 11 % meaning that micro businesses and SME represent 97% of the firms in HCMC. In respect to employment, micro businesses account for 19 % and SMEs for another 25% of the total employment, summing up to 1,3 Mio. Out of 2.9 Mio employees in HCMC.

However, as in many fast-growing countries, official statistics about micro and small businesses in Vietnam in general and in Ho Chi Minh City specifically is limited and fragmented. This implies that the sector is might be still undervalued.

These two paragraph has been added.

Many thanks for the editing suggestions which we all tackle in the revision.

Reviewer 2:

The authors have presented a study micro businesses located in HCMC (Vietnam) concerning their willingness to participate and contribute toward collective flood adaptation measures. This study builds upon existing work while providing a new perspective on the capacity of micro SME to act rather than a stock take of their behaviour/actions. The survey results are presented clearly and is not "surprising" given the knowledge present in the current (limited) body of literature. Overall, I do not have major comments to make. However, I can suggest the following considerations:

We express our gratitude to you for conducting a thorough and positive review of our initial manuscript. Your comment and observations will definitely improve the paper's coherence. Of course, we plan to take them up in our revisions and will provide more clarification.

1) Given that the methodology is in effect a choice experiment? Why was such an approach not considered for this paper formally?

You are right, basically, this study is based on the rationales of discrete choice experiments. However, our research design explicitly emphasizes the following two specifics: To explore the participants individual contribution to a public good, and to explore whether the participant is willing to contribute financially to the public good or free-ride on the others contributions (see e.g., Ones and Putterman 2007). The rationale behind such contributions is called the voluntary contribution mechanism. In our case study, flood adaptation is further defined as a so-called discrete public good, whose provision is only guaranteed if several actors cooperate and individually provide certain

financial contributions (i.e., threshold value). Hence, the public-good game implies that if the provision point of the adaptation measure is not reached, then a money-back guarantee is applied. In this case, the individual contribution of each actor in the experiment is refunded. In Behavioral economics, these underlying rationales and mechanisms are applied in so-called public good games. Overall, they try to explain why collective actions variously succeed or fail. Given our specific research design, we therefore focused on public good games as the complementary theoretical background.

We added this explanation..

2) This is in effect a stated preference study (i.e., you ask the respondents to state if they would or would want a hypothetical outcome), what steps were taken to minimize the potential hypocritical bias in the respondents choices? While hypothetical bias cannot be eliminated in a stated preference study, did you take any steps to limit the potential degree of hypothetical bias in terms of the construction of the scenarios or research processing? I understand that you mention external validity, which is true for the transference of the results but hypothetical bias would have possibly introduced an upwards bias in responses.

Many thanks for this very valuable comment. Here our answer to concerns raised by your comment which will elaborate further in our revision.

We spent a lot of time in advance to make the scenarios as realistic as possible by visiting and interviewing flood-affected businesses, which were then discussed with local experts. We also conducted a qualitative survey of business owners about the actual impacts and adaptation measures. This helped us minimize hypothetical bias, although it cannot be completely eliminated.

We added this explanation.